# A Tribological and Ion Released Research of Ti-Materials for Medical Devices

**DOI:** 10.3390/ma15010131

**Published:** 2021-12-24

**Authors:** Daniela Silva, Camila Arcos, Cecilia Montero, Carolina Guerra, Carola Martínez, Xuejie Li, Armelle Ringuedé, Michel Cassir, Kevin Ogle, Danny Guzmán, Claudio Aguilar, Maritza Páez, Mamié Sancy

**Affiliations:** 1Departamento de Ingeniería Mecánica y Metalúrgica, Escuela de Ingeniería, Pontificia Universidad Católica de Chile, Santiago 7820436, Chile; caguerra2@uc.cl; 2Departamento de Ingeniería Metalúrgica, Facultad de Ingeniería, Universidad de Santiago, Santiago 9170022, Chile; marta.montero@usach.cl; 3Departamento de Ingeniería de Obras Civiles, Universidad de La Frontera, Temuco 4780000, Chile; carola.martinez@ufrontera.cl; 4CNRS, Institut de Recherche de Chimie de Paris, Chimie ParisTech, PSL University, 75005 Paris, France; xuejie.li@chimieparistech.psl.eu (X.L.); armelle.ringuede@chimie-paristech.fr (A.R.); michel.cassir@chimieparistech.psl.eu (M.C.); kevin.ogle@chimieparistech.psl.eu (K.O.); 5Departamento de Ingeniería en Metalurgia, Universidad de Atacama, Copiapó 1530000, Chile; danny.guzman@uda.cl; 6Departamento de Ingeniería Metalúrgica y de Materiales, Universidad Técnica Federico Santa María, Valparaíso 2390123, Chile; claudio.aguilar@usm.cl; 7Departamento de Química de los Materiales, Facultad de Química y Biología, Universidad de Santiago de Chile, Santiago 9170022, Chile; maritza.paez@usach.cl; 8Escuela de Construcción Civil, Pontificia Universidad Católica de Chile, Santiago 7820436, Chile; mamiesancy@gmail.com; 9Centro de Investigación en Nanotecnologiía y Materiales Avanzados “CIEN-UC”, Pontificia Universidad Católica de Chile, Santiago 7820436, Chile

**Keywords:** Ti-6Al-4V, surface characterization, ion release, scratch test

## Abstract

The increase in longevity worldwide has intensified the use of different types of prostheses for the human body, such as those used in dental work as well as in hip and knee replacements. Currently, Ti-6Al-4V is widely used as a joint implant due to its good mechanical properties and durability. However, studies have revealed that this alloy can release metal ions or particles harmful to human health. The mechanisms are not well understood yet and may involve wear and/or corrosion. Therefore, in this work, commercial pure titanium and a Ti-6Al-4V alloy were investigated before and after being exposed to a simulated biological fluid through tribological tests, surface analysis, and ionic dissolution characterization by ICP-AES. Before exposure, X-ray diffraction and optical microscopy revealed equiaxed α-Ti in both materials and β-Ti in Ti-6Al-4V. Scratch tests exhibited a lower coefficient of friction for Ti-6Al-4V alloy than commercially pure titanium. After exposure, X-ray photoelectron spectroscopy and surface-enhanced Raman spectroscopy results showed an oxide film formed by TiO_2_, both in commercially pure titanium and in Ti-6Al-4V, and by TiO and Al_2_O_3_ associated with the presence of the alloys. Furthermore, inductively coupled plasma atomic emission spectroscopy revealed that aluminum was the main ion released for Ti-6Al-4V, giving negligible values for the other metal ions.

## 1. Introduction

Commercially pure titanium (CP-Ti), and its alloys, have been extensively investigated due to their use in the manufacture of orthopedic devices. Their properties, such as low density, high specific resistance, and good corrosion resistance, make these materials attractive for that particular application [1,2]. Specifically, the Ti-6Al-4V alloy is used in dental and orthopedic implants [3,4]. However, the exposure of this alloy to biofluids has been shown to foster ion release [5,6], promoting harmful and undesirable effects for the human body [7], ranging from the induction of allergies to the promotion of the granuloma’s formation and even carcinomas [6,8]. Concerning the harmful effects, genotoxicity was associated with vanadium release from in vitro studies up to peripheral neuropathy and even Alzheimer’s disease [9,10,11]. Aluminum is also considered a neurotoxin, increasing the risk of degenerative diseases such as Alzheimer’s [12,13,14]. Therefore, using these materials in biomedical applications require the guarantee of no potential adverse effects of released metal cations during their exposure to biofluids.

A method for quantifying ion release in situ is through atomic emission spectroelectrochemistry (AESEC), which has been successfully applied to monitor the ion release from metallic implants at laboratory scale [15,16,17]. In this sense, Luxin et al. [18] analyzed Mg^2+^ release from Ti-xMg (x = 0.312, 0.625, 1.25 and 2.5 wt.%) after 1, 3, 7, 14, and 28 days of immersion at 37 °C. The authors quantified ion release of all alloys, which exhibited similar behavior, and observed a quick release the first day followed by an increase in longer exposure times. Furthermore, the authors found that the cation release rate at each time depended on the Mg content; thus, the maximum release followed this trend: Ti-0.312 Mg < Ti-0.625 Mg < Ti-1.25 Mg < Ti-2.5 Mg. Guerra et al. [19] also used this technique to observe the ion release in a porous Ti-20Nb-11Ta-16Fe-1Mn after 30 days of exposure to simulated body fluid (SBF). No ion proceeding from the alloy was quantifiable by AESEC but inductively coupled plasma atomic emission spectrometry (ICP-AES) enabled detection of 1.9 and 4.8 ppb of Mn and Fe, respectively, attributed to an increase in the thickness of the oxide layer during the exposure, improving passivation. Yan et al. [20] investigated the effect of Zr ion release for coated-Ti with Zr ion-implantation on osteogenesis. These authors found that more particles were released for longer ion implantation time, which was related to an increase in genes due to osteogenesis. On the other hand, Yamazoe et al. [21] determined that the release of Ti cations is related to microstructure, where the level of release decreased when the titanium grain size was small for titanium/dental alloy. Hu et al. [22] also observed that Ti cation release diminished with grain size decreased in Ti-45Nb when they compared a coarse-grained sample by casting with an ultrafine-grained (UFG) sample by high-pressure torsion. The authors explained that the UFG sample promotes a thicker surface passive layer with fewer defects.

Moreover, it has been reported that wear debris is a significant cause of periprosthetic osteolysis, which corresponds to a wear product [23]. Eger et al. [24] investigated the effect of surface treatment of the Ti disc on particle release associated with inflammation and osteolysis. In comparison with sandblasted/acid-etched material, sandblasted Ti showed a more significant amount of particle release, possibly due to a significant occurrence of protein-related inflammation and a critical bone loss for in vivo assays. Additionally, Zhang et al. [25] analyzed this using a natural analgesic and anti-inflammation drug (Bulleyaconitine or BLA) to prevent osteolysis induced by Ti particles. When Ti particles in contact with the bone were embedded with BLA, bone resorption decreased by suppressing osteoclast formation and stimulating osteoblast differentiation and mineralization. Therefore, the surface condition of Ti-6Al-4V plays an important role when estimating what phenomena an implant will undergo within the body.

Therefore, in this work, to better understand the corrosion and/or wear mechanisms of commercially pure titanium (CP-Ti) and Ti-6Al-4V, a comparison of both materials was carried out by using tribological, surface, and electrochemical analyses [26,27], studying mainly the surface damage.

## 2. Materials and Methods

### 2.1. Samples

CP-Ti and commercial Ti-6Al-4V wt.% were purchased (Chilexpo) as wrought samples having a 25 mm diameter, which were mechanized to achieve 8 mm and 6 mm diameter and height, respectively. For CP-Ti, the chemical composition was (in wt.%): 0.02 C, 0.015 H, 0.03 Fe, 0.03 N, 0.25 O, and balance Ti. For Ti-6Al-4V, the chemical composition was (in wt.%): 0.08 C, 0.015 H, 0.25 Fe, 0.05 N, 0.20 O, 5.5–6.75 Al, 3.5–4.5 V, and balance Ti. Both samples were mechanically polished using, first, SiC paper from 600 to 2400 grade, then, with a colloidal silica suspension of 0.05 μm. Later, the samples were cleaned with an ethanol solution in a sonicator bath for 10 min.

### 2.2. Structural and Microstructural Characterization

X-ray diffraction (XRD) patterns of CP-Ti and Ti-6Al-4V samples were also collected with a Shimadzu XDR-6000 diffractometer with Cu-Kα radiation (Shimadzu Corp., Tokyo, Japan), with an angular step of 0.02° (2θ) and a time for a step of 6 s. Furthermore, CP-Ti and Ti-6Al-4V samples were etched with Kroll’s reagent for 20 s to observe the microstructure using an Olympus model-BX46M optical microscope (Olympus Corporation, Shinjuku-ku, Tokyo, Japan) for microstructural analysis.

### 2.3. Tribological Characterization

Scratch tests were performed using the single-pass. The coefficient of friction (COF) was estimated by applying 1 N and 5 N loads at room temperature and dry sliding conditions, with a tangential scratch speed of 2.5 mm s^−1^, a scratch length of 5 mm, and in triplicate, as described previously by Martínez et al. [28]. A tip stylus with a Rockwell-C diamond cone with a 120° opening angle and a 200 µm spherical tip radius and a linear drive indenter (UMT2-Bruker) was used, and the wear track volume (DV) analysis was performed using a 3D-profilometer (CCI-MP-3D Taylor-Hobson). In this case, CP-Ti and Ti-6Al-4V samples were also observe using an Olympus model-BX46M optical microscope (Olympus Corporation, Shinjuku-ku, Tokyo, Japan).

### 2.4. Exposure

CP-Ti and Ti-6Al-4V samples were exposed to 50 mL of naturally aerated Hank’s solution using a thermostatic bath operated at 37 °C. Hank’s solution contained (in g/L): 0.14 CaCl_2_, 0.40 KCl, 8.00 NaCl, 0.10 MgSO_4_, 0.06 KH_2_PO_4_, 0.05 NaHPO_4_, and 1.00 glucose [29]. The exposure was performed in a static condition, and Hank’s solution was refreshed partially after 7 days of exposure.

### 2.5. Surface Analysis

An X-ray photoelectron spectrometer FlexPS SPECS with an energy scale (XPS, SPECS Surface Nano Analysis GmbH Voltastrasse 5, Berlin, Germany) calibrated with the binding energy of advantageous hydrocarbons (C-C/C-H) in the C1s signal at 285.2 eV was used. The fitting and decomposition of the curve were performed after removing the Shirley-type background. Before and after exposure, Raman spectra were collected using a Renishaw InVia confocal Raman microscope with WiRE software and a green laser (doubled Nd: YAG, 532 nm) with 10% laser power (Renishaw, New Mills, Wotton under Edge, UK). Raman measurements were performed with Leica objective (×50, NA = 0.75). In this case, the exposure time of the laser was 1 s, with 30–50 acquisitions and an estimated 0.87 µm laser spot diameter. XPS and Raman after exposure were realized after 1 and 14 days, respectively. Morphological characterization was performed using a field emission-scanning electron microscope (FE-SEM, Quanta FEG 250, Thermo Fisher Scientific, Waltham, MA, USA) with secondary electrons. Additionally, chemical composition analysis of samples was carried out using an energy-dispersive spectrometer (EDS, Thermo Fisher Scientific, Waltham, MA, USA) that used a silicon drift detector (SDD, Thermo Fisher Scientific, Waltham, MA, USA), with an Octane Pro model with 10 mm^2^ detection area of 130 eV resolution, and resolution stability over 90% up to 200 kcps (Thermo Fisher Scientific, Waltham, MA, USA).

### 2.6. Ionic Dissolution Characterization

Real-time dissolution analysis of CP-Ti and Ti-6Al-4V samples was performed using AESEC. This technique consisted of an electrochemical flow cell coupled to an inductively coupled plasma atomic emission spectrometer (ICP-AES) (Ultima 2C Horiba-Jobin Yvon, Palaiseau, France) for downstream elemental analysis. A three-electrode flow cell was used, which was constructed in-house, as desbribed by K. Ogle [30]. A saturated calomel electrode and platinum wire were employed as a reference and counter electrode, respectively. A flow rate of 2.86 mL min^−1^ was passed through the cell, and electrochemical (E, i) and spectroscopic emission data were collected with a time resolution of 1 data per second. The emission wavelengths used were 337.279 nm for Ti, 396.152 nm for Al, and 311.071 nm for V and were calibrated by the standard methods using commercial standards of the elements in Hank’s solution electrolyte. The detection limit (DL), in parts per billion (ppb) was 1.3 for Ti, 18 for Al, and 2.7 for V. The detection limit was defined as DL = 3s, where s is the standard deviation of the background signal of the blank solution. Elemental dissolution was monitored after 14 days of exposure at 37 ± 1 °C during an electrochemical sequence an open circuit potential (EOC), followed by linear sweep voltammetry (LSV) at a scan rate of 0.25 mV s^−1^. After that, alternating EOC and overpotentials (η) conditions were applied, and a chronoamperometry at η = 0.5 V and η = 1 V was also applied.

Additionally, the accumulated elemental dissolution was measured during the exposure described in Section 2.4 by removing a 20 mL aliquot of the exposed electrolyte after 7 and 14 days of exposure to Hank’s solution and subjecting the aliquot to elemental analysis by ICP-AES (Ultima 2C Horiba-Jobin Yvon, Palaiseau, France). After extracting the aliquot, the immersed solution was completed with a fresh 20 mL solution.

## 3. Results and Discussion

### 3.1. Structural and Microstructural Analysis

Figure 1 shows the XRD patterns of CP-Ti and Ti-6Al-4V samples before exposure. CP-Ti peaks correspond to hexagonal α-Ti phase, while Ti-6Al-4V peaks are associated with α-Ti and cubic β-Ti phases. These are the typical phases in the Ti-6Al-4V alloy heat-treated with a low cooling rate [31,32]. It should be noticed that the presence of alloying elements modifies the microstructure from a pure α-Ti phase to (α+β)-Ti phases [33,34]. The principal difference is the appearance of the peaks in Ti-6Al-4V around 37.5°, 57°, and 71°, possibly associated with the β-Ti phase due to the presence of β-Ti stabilizer, vanadium [35,36,37]. Additionally, the relative intensity of the peak at 38°, corresponding to the ($10\bar{1}0$) plane [38], is lower for Ti-6Al-4V than CP-Ti. This can be associated with a prismatic texture of Ti-6Al-4V, where a higher plastic deformation results in lower peak intensity [39,40,41]. This influence is also observed at 53°, 63°, 71° and 77°, which can be related to heat treatment, as described by Beladi et al. [42].

Figure 2 shows the microstructure of CP-Ti and Ti-6Al-4V before exposure. Figure 2a shows the CP-Ti microstructure with an equiaxial α-Ti, as was previously reported by Amanov et al. [43], Gil et al. [44], and Greger [45]. It is possible that darker zones were generated by over-etching, as was proposed by G. Vander Voort [46]. A matrix characterizes Ti-6Al-4V microstructure with a phase homogeneously distributed, attributed to equiaxial α-Ti and intergranular β-Ti (lighter and darker zones in Figure 2b), respectively, as reported by Çaha et al. [47] for commercial Ti-6Al-4V, and by Dabrowski [48] for rolled and annealed Ti-6Al-4V at low cooling rates.

### 3.2. Tribological Characterization

Figure 3 shows the COF of CP-Ti and Ti-6Al-4V at two applied loads, 1 and 5 N. The COF increased during the scratch test for a load of 1 N (see Figure 3a), also revealing an oscillatory behavior for the CP-Ti sample that can be attributed to the susceptibility to the friction-induced instability, possibly due to the material roughness (not determined here). The COF amplitude variation is called the stick-slip effect [49,50], which agrees with Amonov et al. [43], who suggested that this variation could be attributed to the localized fracture of the transfer layer and the interaction of the particles at the sliding interface. For 1 N load, the average values of COF were close to 0.63±0.11 for CP-Ti and 0.65 ± 0.09 for Ti-6Al-4V. Instead, for 5 N load, the COF was 0.64 ± 0.06 for CP-Ti, being reduced to 0.46 ± 0.04 for Ti-6Al-4V. Therefore, the COF decreases in the Ti-6Al-4V sample when the applied load increases, agreeing with Gain [51] and Yazdi [52]. This reduction can be associated with the higher coverage of the tribolayer on the wear surface with higher normal loads, which decreases the adhesion between mating surfaces, as proposed by Yazdi et al. [52]. Li et al. [53] observed a decrease in the coefficient of friction at higher loads at a different sliding velocity in Ti-6Al-4V, which was related to an increase of the hardness with the load.

Figure 4 shows the indenter penetration depth for wear tracks of CP-Ti and Ti-6Al-4V samples. According to Vencl et al. [54], the scratch test by a single point is used to study the interaction that occurs between the abrasive particle (tip) and the surfaces to be studied (CP-Ti and Ti-6Al-4V), where the abrasive particle (tip) simulates wear.

The grooved surface of the CP-Ti sample had a penetration value close to −0.7 μm of depth (see Figure 4a), while the profile of scratch of the Ti-6Al-4V sample had a penetration value relative to −0.3 μm of depth (see Figure 4b). This behavior has been attributed to the hardness difference, close to 146 HV_0.5_ for CP-Ti [43] and 350 ± 5 HV_0.5_ for Ti-6Al-4V [47] by solution hardening of Al in Ti-6Al-4V. The phases volume fraction also influences hardness, as was studied by Patil et al. [55], in Ti-6Al-4V with different thermal treatment, where hardness was reduced with an increase of alpha volume fraction. This is according to Archard’s law, where a higher hardness is related to lower wear volume with a constant load [56]. On the other hand, the area of the groove surface of CP-Ti was near 51.1%, and the pile-up area was 48.8%. Therefore, 99.9% of the groove surface transferred to the pile-up, which means that 0.07% corresponded to a loss of material. In accordance with Rajendhran et al. [57], this could be due to the micro-ploughing mechanism, which represents the steady-state formation of a groove that has ridges on both sides formed by local plastic deformation without the formation of wear debris because the percentage of material loss is almost negligible. The area of the groove surface of Ti-6Al-4V was close to 71.8%, and the pile-up area 27.9%. Therefore, 38.9% of the groove surface transfer to the pile-up, which means that 61.1% corresponds to a loss of material. A dry sliding wear study made by Alam et al. [58] on Ti-6Al-4V showed a severe adhesion to the counter body, which could explain the significant amount of lost material.

As reported by Rajendhran et al. [57], wear in terms of scratch width and depth are the possible means to measure the damage assessment. In most cases, the scratch width is estimated to control the wear. However, the width of the cross-section profile scratches is similar for the surface of CP-Ti and Ti-6Al-4V, which have values of 72.19 μm and 66.82 μm. While the depth or volume loss is more important for the wear resistance application, therefore, the shallowest depth corresponds to the Ti-6Al-4V surface, with an approximate value of −0.3 μm, which would correspond to a greater resistance to wear compared to the CP-Ti surface with a depth value close to −0.7 μm.

Consequently, the values of groove surface penetration indicate that the plastic deformation of CP-Ti is higher than Ti-6Al-4V. Farokhazes et al. [59] studied the surface of Ti-6Al-4V under a nitrogen atmosphere. According to the profile, they performed the same scratch test in a single pass but with an applied force of 10 N, giving a deformation of about −2.5 µm. This value is almost 10 times greater than that reported in this survey with an applied force of 5 N, despite the similar microstructure (equiaxial α-Ti and intergranular β-Ti). This difference could be due to nitrides on the Ti-6Al-4V surface subjected to a nitrogenous environment and the higher hardness (430 ± 8 HV) of Farokhazes’s sample.

Figure 5 shows the wear tracks’ morphology to determine the specimens’ wear mechanism subjected to different loads on the CP-Ti and Ti-6Al-4V samples. As mentioned above, samples were tested with two different loads, 1 N (see Figure 5a,b) and 5 N (see Figure 5c,d). For 1 N load, the surface of CP-Ti exhibited a noticeable plastic deformation that generated the accumulated material in different areas (red circle), micro-cracking in the edges of the wear tracks (yellow circle), and debris (black arrow). However, the mechanism changed for 5 N possibly due to the more homogeneously drag of accumulated material on the surface but the noticeable plastic deformation (red circle) and micro-cracking stayed in the edges of the wear tracks (yellow circle), revealing delamination as a new mechanism (orange circle). This behavior of the surface is in agreement with the literature, which stipulated that the dry contact involves different wear mechanisms that emerge at the same time [51,60,61]. For example, Gain et al. [51] found a severe adhesion between Ti-alloys and its counterpart during the transfer of Ti-6Al-4V dry contact sliding. Also, Li et al. [60] reported grooves and cracks on the Ti-6Al-4V surface after wear. In this context, Yildiz et al. [61] proposed that the wear particles first appear due to abrasion in the surface of Ti-6Al-4V, and were added to the pin later, increasing the friction. Additionally, it should be noticed that after 1 N load was applied (Figure 5b), the surface of Ti-6Al-4V exhibits ploughing (white arrow) and, in some areas, delamination (orange circle), as shown in Figure 5b. Nevertheless, when the load increases to 5 N, deep ploughing (pink arrow) and the mechanisms mentioned before were observed, as proposed by Gain et al. [51], who also reported severe abrasive wear.

### 3.3. Surface Analysis

Figure 6 shows the Ti-6Al-4V XPS survey spectra and high-resolution spectra of Ti 2p, O 1s, C 1s after 1 day of exposure to Hank’s solution. This work used carbon as internal reference at 285.1 and 288.3 eV for calibrating peak positions. The Ti 2p spectra revealed two doublets at 455.8 eV and 464.3 eV associated with Ti^2+^ and Ti^4+^, respectively [62,63,64]. Mendis et al. [65] analyzed the oxide film formed on Ti-xTa alloys (x = 10, 20, 30, 50, 60, and 75 wt%.) after exposure to Hank’s solution, also observing two doublets, but at 453.61 eV for TiO and 458.54 eV for Ti^4+^, which were related to metallic titanium and TiO_2_. In this case, the authors also observed signals at 284.8 eV, 286.24 eV, and 288.8 eV attributed to the presence of C–C/C–H, C–O, and C=O, possibly due to environmental contamination. Xu et al. [66] studied the oxide layer of Ti-6Al-4V and Ti-25Nb-10Ta-1Zr-0.2Fe after exposure to Ringer’s solution at 37 °C, observing three doublets for Ti^4+^ (464.3 eV), Ti^3+^ (462.3 eV), and Ti^2+^ (455.4 eV) that were related to TiO_2_, Ti_2_O_3_, and TiO. Furthermore, the doublet areas were calculated, revealing that TiO_2_ was present with 78%, Ti_2_O_3_ with 15%, and TiO with 7%. The same doublets were found at the Ti 2p spectrum of Ti-25Nb-10Ta-1Zr-0.2Fe but slightly shifted. The relative contents calculated for each doublet indicated that TiO_2_ was close to 83%, Ti_2_O_3_ was 13%, and TiO was near 4%. They agree with the relative contents calculated of Ti species in this work, where TiO_2_ had the most significant presence, showing 93.5% and 6.5% for TiO. Deconvolution of O 1s spectra revealed three contributions at 528.9 eV, 530.1 eV, and 532.1 eV corresponding to O^2−^, Ti-O [67,68] and O from absorbed water [69], respectively. Similar behavior was observed by Silva et al. [29] for a porous Ti-6Al-4V sample that was exposed to Hank’s solution at 37 °C. The O 1 s spectrum revealed the presence of the O^2−^ signal at 529.9 eV, associated with Al_2_O_3_. The following signals were also found: Ti-O at 530.4 eV, O-H at 531.7 eV in O 1s, and Ti^4+^ at 464.8 eV, Ti^3+^ at 459.8 eV and Ti^2+^ at 454.4 eV.

Figure 7 shows the Raman spectra of CP-Ti and Ti-6Al-4V before and after 14 days of exposure to Hank’s solution, which revealed Raman bands in agreement with the active Raman modes of TiO_2_ that have been previously reported between 100–900 cm^−1^ [70,71]. Before exposure, the spectra of CP-Ti revealed bands at 150.9 cm^−1^, 224 cm^−1^, 803.4 cm^−1^, while after exposure, the bands were slightly shifted at 146.6 cm^−1^, 232.9 cm^−1^, and 797.7 cm^−1^. Ekoi et al. [72] studied the effect of a microwave plasma process on the CP-Ti disc to promote an oxide layer growing before exposure to an oxygen atmosphere. The authors registered the Raman bands at 143.2 cm^−1^, 446.6 cm^−1^, 609.8 cm^−1^, and 800.27 cm^−1^, which were related to B_1g_, E_g_, A_1g_, and B_2g_ Raman active modes. In this case, B_1g_ was attributed to symmetric bending vibration, E_g_ to symmetric stretching vibration, and A_1g_ to anti-symmetric bending vibration of O–Ti–O [73,74]. For Ti-6Al-4V, similar bands were observed at 152 cm^−1^, 222.9 cm^−1^, 798.9 cm^−1^ before exposure, and at 143.2 cm^−1^, 233.0 cm^−1^ and 802.3 cm^−1^ after exposure. Furthermore, Nouicer et al. [75] also exposed the Ti-6Al-4V to an SBF solution, observing that Raman bands were close to 150 cm^−1^, 230 cm^−1^, and 800 cm^−1^, attributed to TiO_2_ (rutile). Shaikh et al. [76] studied the bioactivity and biocompatibility of Ti-6Al-4V after a surface modification by using a laser technique, which revealed Raman bands at 151 cm^−1^, 217 cm^−1^, 417 cm^−1^, and 610 cm^−1^ after modification that were associated mainly with TiO_2_ and Ti_3_O_5_, as oxide. Therefore, XPS and Raman analyses revealed that the oxide layer formed on the CP-Ti and Ti-6Al-4V was mainly formed by TiO_2_, with a minor contribution of Al_2_O_3_ for Ti-6Al-4V.

Figure 8 shows the FE-SEM images for CP-Ti and Ti-6Al-4V before and after 14 days of exposure and under an overpotential condition. Before exposure, no differences were observed between the samples, without superficial imperfections. However, after 14 days of exposure, the CP-Ti sample did reveal a higher surface damage than that of the Ti-6Al-4V sample, agreeing with the higher corrosion rate for CP-Ti than Ti alloys, as reported by Gurappa [77]. In fact, in both images, it is possible to observe scratch lines resulting from the polishing process, which can be associated with a more stable and harder passive oxide layer over Ti-6Al-4V. According to Lee et al. [78,79], in commercial Ti and Ti-6Al-4V specimens, a distinct structural change is not observed on the surface even after samples being soaked in Hank’s solution for 30 days.

### 3.4. Real-Time Elemental Dissolution Analysis

Figure 9 shows the AESEC results of CP-Ti and Ti-6Al-4V after 14 days of exposure to Hank’s solution at 37 °C. Notice that the electrochemical data and the simultaneous atomic emission measurement for the respective elements can be seen in Figure 9. The emission intensities (I_λ_) as a function of time show no elemental dissolution during the sequence of electrochemical experiments. The detection limits are shown on the right-hand side of the respective curves.

It should be noted that I_λ_ was corrected for background drift by normalizing with the emission intensity at 371.029 nm, associated with the element Y, which is not present in our system. Failure to make this correction could falsely attribute slow changes in the emission intensity background to changes in the elemental dissolution rate. The horizontal dashed lines represent the separation of 3 s, which is commonly used to define the detection limit for a given element and wavelength. The value of the detection limit expressed as an equivalent current is also given on the right. All the signals remain within the 3 s limit indicating that no measurable dissolution was observed. For CP-Ti, the electrical current remained lower than the detection limit of Ti, expressed as an equivalent current, throughout the experiment except for the +1 V potentiostatic experiment. The measurement of ionic dissolution would not be expected for these experiments under any circumstance. However, for the +1 V potentiostatic experiment, the electrical current is more than one order of magnitude above the detection limit. The absence of detectable ionic dissolution during this step suggests that it makes a negligible contribution during the anodic oxidation of the material. The Ti-6Al-4V also showed no detectable ionic dissolution even though the electrical current was above the detection limit of Ti for each of the three dynamic experiments performed. The dissolution of the CP-Ti and Ti-based alloy was investigated for longer times by removing a 20 mL aliquot of the electrolyte during the exposure described in Section 2.4, after 7 and 14 days. The accumulated concentration of Ti, V, and Al as determined by ICP-AES, as shown in Table 1. Ti and V were consistently below the detection limit (DL). For Ti-6Al-4V, however, a slight dissolution of Al was detected, rising to about 5.37 µg·cm^−2^ to 6.96 µg·cm^−2^ after 7 days to 14 days.

Similar behavior was reported by Guerra et al. [19], who studied the porous alloy Ti–Nb–Ta–Fe–Mn after 30 days immersed in Ringer’s solution at 37 °C. These authors also failed to detect release even after applying η = +1 V against SCE, which was associated with the formation and stability of the oxide formed and possibly, with good passivity, stable even at high overpotentials. According to the Raman analysis, the oxide film for both samples was composed mainly of TiO_2_ as rutile, which is a thermodynamically stable oxide [80] in standard conditions. It should also be noticed that the rutile exhibits a low Gibbs energy-free formation [81,82] that should be stable even after applying a η = +1 V. Table 1 also shows ICP-AES results of CP-Ti, which exhibited a concentration of Ti under the DL for all exposure times.

On the other hand, Ti-6Al-4V alloy revealed a release in Al ion after 7 days of exposure, increasing after 14 days. A similar result was previously reported by Prodana et al. [83]. The authors reported ion release of all alloying elements, including Al, for shorter exposure times and decreased ion releases after 14 days. This phenomenon is expected because the α-Ti phase is rich in Al, preferably dissolved [84,85,86]. In this context, Chen et al. [83] investigated Ti-6Al-4V exposed to simulated acidic solutions (pH 3 and 5), observing a preferential dissolution of the α-Ti over β-Ti by the optical surface profiler technique. It should also be mentioned that XPS (see Figure 6) revealed peaks related to Al_2_O_3_, indicating that Al can also be part of the passive film.

Moreover, it has been observed that the alloys can suffer significant changes during the earlier stages, as was described by Silva et al. [29], who studied the Ti-6Al-4V exposed to Hank’s solution for 21 days. Wang et al. [87] detected elemental Ti dissolution from CP-Ti under open circuit and polarization conditions in a similar electrolyte (minimum equivalent media, or MEM), determining that the reactivity enhances with the addition of 0.1% albumin and 0.1% H_2_O_2_. Time-resolved ICP mass spectrometry showed the equivalent faradaic dissolution rate reached approximately 4 µA cm^−2^ when the electrical current was ≈ 8 µA cm^−2^, during a polarization using a similar electrochemical flow cell. It means a 50% faradaic yield of dissolution, demonstrating the aggressiveness of the modified electrolyte.

## 4. Conclusions

The characterization of the wear and corrosion behavior of CP-Ti and Ti–Al–V, using COF measurements and quantification of released ions, revealed that Ti–Al–V has a better performance. This was evidenced by the wear properties, where a lower COF and lower wear residue than CP-Ti were observed. This is partly explained by the RAMAN, XPS, and XRD results, which revealed that the oxide film formed on Ti–Al–V surface is composed mainly by TiO_2_ and Al_2_O_3_. However, the structural analysis revealed that alloy is composed of β-Ti and α-Ti phases, which have a better mechanical respond than α-Ti that is present in CP-Ti.

On the other hand, the quantification of released ions showed that Ti–Al–V exposed to Hank’s solution has a robust barrier against corrosion, observing a low activity of released ions compared to CP-Ti. This is related to the passive oxide film formed on the surface, as verified by Raman and XPS, which protects the material from hasty dissolution. In fact, ICP-AES demonstrates that when an anodic overpotential was applied to promote the anodic reaction, Ti–Al–V showed low quantification of released ions for Ti and V, being slightly higher for Al.

## Figures and Tables

**Figure 1 materials-15-00131-f001:**
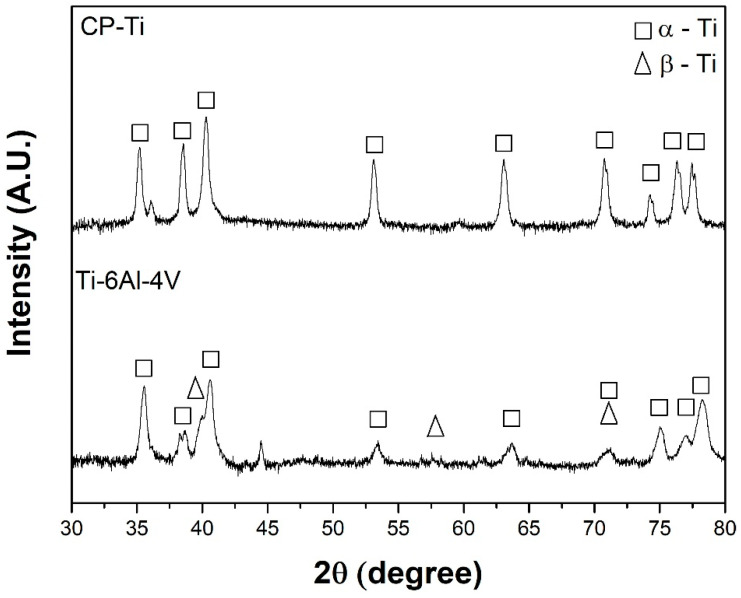
X-ray diffraction (XRD) patterns of CP-Ti and Ti-6Al-4V before exposure.

**Figure 2 materials-15-00131-f002:**
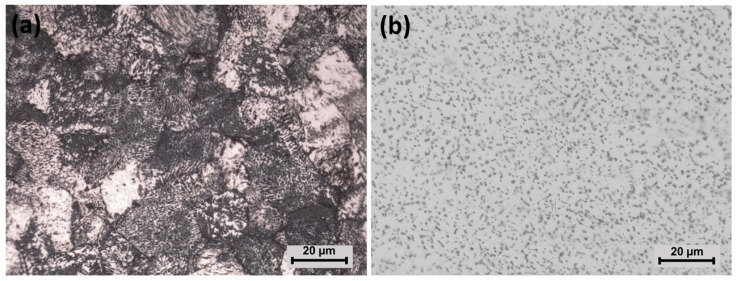
Optical images of (**a**) CP-Ti and (**b**) Ti-6Al-4V prior to exposure.

**Figure 3 materials-15-00131-f003:**
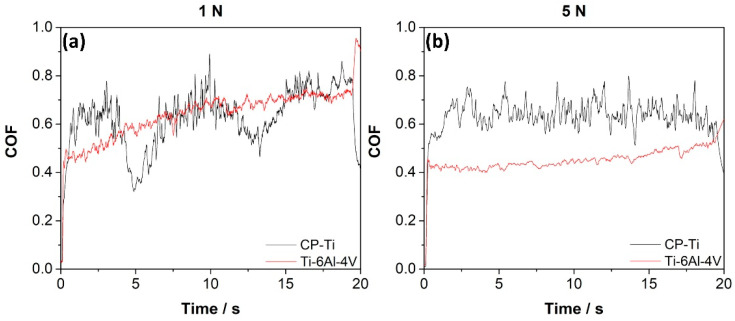
Coefficient of friction (COF) variation of the CP-Ti and Ti-6Al-4V surfaces under load of (**a**) 1 N and (**b**) 5 N.

**Figure 4 materials-15-00131-f004:**
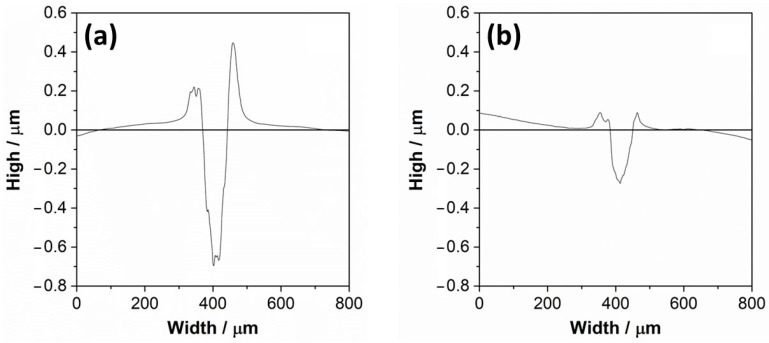
Cross-section profiles of scratches on the surface of (**a**) CP-Ti and (**b**) Ti-6Al-4V.

**Figure 5 materials-15-00131-f005:**
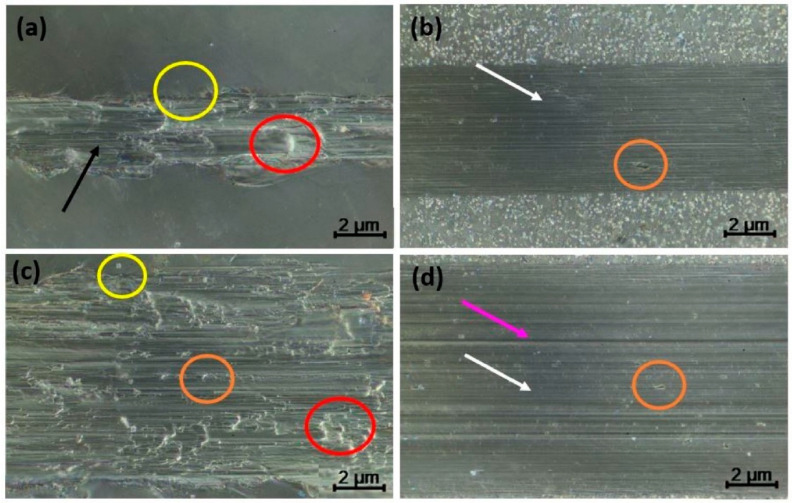
Optical micrographs of the wear track of CP-Ti (**a**,**c**) and Ti-6Al-4V (**b**,**d**) under 1 N (**a**,**b**) and 5 N (**c**,**d**).

**Figure 6 materials-15-00131-f006:**
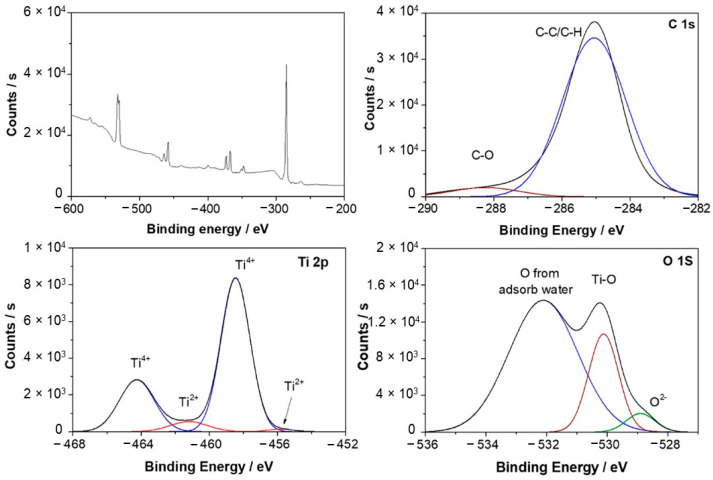
X-ray photoelectrom spectrometry (XPS) survey and C1 s, Ti 2p and O 1 s spectra of Ti-6Al-4V after 1-day exposure in simulated body fluid.

**Figure 7 materials-15-00131-f007:**
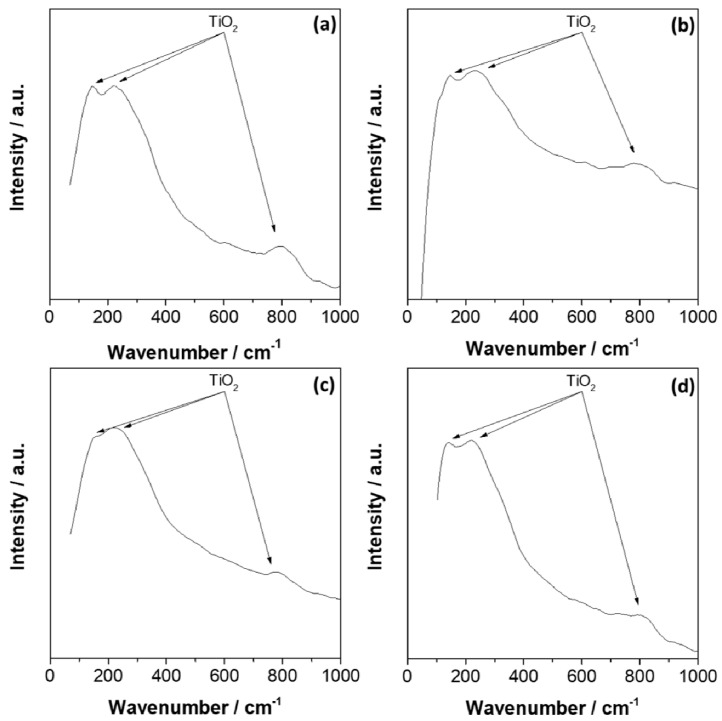
Raman spectra of pure Ti (**a**,**b**) and Ti-6Al-4V (**c**,**d**) before (**a**,**c**) and after (**b**,**d**) 14 days of exposure to Hank’s solution.

**Figure 8 materials-15-00131-f008:**
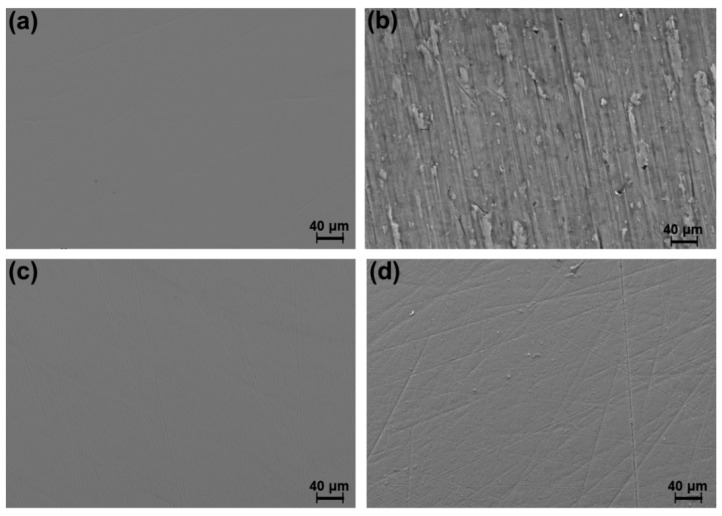
Scanning electron microscope (SEM) images of (**a**,**b**) CP-Ti (**c**,**d**) Ti-6Al-4V (**a**,**c**) prior to and (**b**,**d**) after 14 days of exposure to Hank’s solution.

**Figure 9 materials-15-00131-f009:**
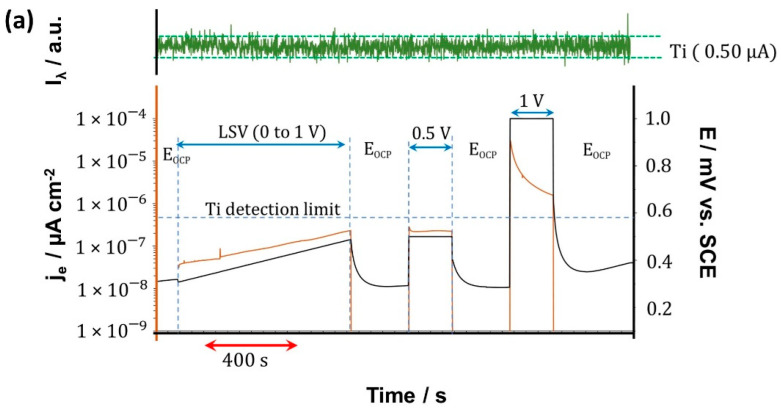
Atomic emission spectroelectrochemistry (AESEC) results of (**a**) Ti and (**b**) Ti-6Al-4V after 14 days of exposure to Hank’s solution.

**Table 1 materials-15-00131-t001:** Accumulated elemental dissolution measured for CP-Ti and Ti-6Al-4V samples after 7 and 14 days of exposure to Hank’s solution.

ICP-AES	Accumulated Mass per Area/µg·cm^−2^			
	CP-Ti	Ti-6Al-4V		
Time of Exposure/days	Ti	Ti	Al	V
7	<DL	<DL	5.37	<DL
14	<DL	<DL	6.96	<DL

## Data Availability

The raw/processed data required to reproduce these findings cannot be shared at this time due to technical or time limitations.

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
