# Peer review of "A Tribological and Ion Released Research of Ti-Materials for Medical Devices"

_materials, 2021, doi:10.3390/ma15010131_

Round 1
Reviewer 1 Report
The manuscript entitled: A tribological and ion released research of Ti-materials for medical devices deals with the Ti-based biomedical materials especially w.r.t tribology and ion release research. The topic itself is very important in the field of biomedical research and the results are quite interesting. I have the following concerns with the present manuscript.
- The authors should introduced how did they prepare the CP-Ti and Ti6Al4V samples. Either by casting/PM/wrought?
- The idea of measuring the ion release is understandable. However, the testing of wear/scratch (tribology) needs justification.
- A strong scientific discussion is missing correlating the tribology and ion release w.r.t. the composition.
- The typos in the manuscript should be rectified. For instance, line 113: ....... 2.5 mm s-1 should be written as 2.5 mm s-1.
- The English language needs attention.
Reviewer 2 Report
There are several questions about your research:
1) Hank's solution contains significant amounts of chlorine and sulfur ions. How did you get rid of their influence when determining the concentrations of aluminum, titanium and vanadium in a given solution by the inductively coupled plasma method?
2) Why did you limit yourself to only 14 days of exposure of samples in Hank's solution? Typically, such studies take at least a month.
3) According to Fig. 9, pure titanium was much more attacked in Hank's solution than Ti-6Al-4V alloy. This contradicts the data on the significant dissolution of aluminum. How do you explain this?
Round 2
Reviewer 1 Report
The authors have satisfactorily addressed my comments and the manuscript may now be accepted for publication in the present form.
Author Response
Reviewer 1
The authors have satisfactorily addressed my comments and the manuscript may now be accepted for publication in the present form.
Response: Thank you very much, we appreciate it.
Reviewer 2 Report
As I know, excess chlorine ions affect the determination of the metal ion concentration in ICP AES. This was the question of how accurately the concentrations of titanium, aluminum, and vanadium ions were determined after 14 days of exposure in a high-chloride solution. Did you run blank tests, did you dissolve a known concentration of aluminum ions in Hank's solution to test the performance of the spectrometer?
